# NEURAL DISCRETE REINFORCEMENT LEARNING

## ABSTRACT

Designing effective action spaces for complex environments is a fundamental and challenging problem in reinforcement learning (RL). Some recent works have revealed that naive RL algorithms utilizing well-designed handcrafted discrete action spaces can achieve promising results even when dealing with high-dimensional continuous or hybrid decision-making problems. However, elaborately designing such action spaces requires comprehensive domain knowledge. In this paper, we systemically analyze the advantages of discretization for different action spaces and then propose a unified framework, Neural Discrete Reinforcement Learning (NDRL), to automatically learn how to effectively discretize almost arbitrary action spaces. Specifically, we propose the Action Discretization Variational AutoEncoder (AD-VAE), an action representation learning method that can learn compact latent action spaces while maintain the essential properties of original environments, such as boundary actions and the relationship between different action dimensions. Moreover, we uncover a key issue that parallel optimization of the AD-VAE and online RL agents is often unstable. To address it, we further design several techniques to adapt RL agents to learned action representations, including latent action remapping and ensemble Q-learning. Quantitative experiments and visualization results demonstrate the efficiency and stability of our proposed framework for complex action spaces in various environments.

## 1 INTRODUCTION

Recent advances in Reinforcement Learning have yielded many promising research achievements Vinyals et al. (2019); Berner et al. (2019); Schrittwieser et al. (2019). However, the complexity of action spaces still prevents us from directly utilizing advanced RL algorithms to real-world scenarios, such as high-dimensional continuous control in robot manipulation Lillicrap et al. (2016) and structured hybrid action decision-making in strategy games Kanervisto et al. (2022). Complex action spaces lead to extensive challenges in designs of policy optimization Xiong et al. (2018b), efficiency of exploration Seyde et al. (2021b) and behaviour stability of learned agents Bester et al. (2019).

To handle these issues, some existing work first elaborately design particular reinforcement learning methods in original complex action spaces. Specifically, deterministic policy gradient methods Lillicrap et al. (2016); Fujimoto et al. (2018) are designed to handle continuous control problems. And Xiong et al. (2018b); Fan et al. (2019b) propose some techniques to extract the relationship between different action dimensions, which is important in hybrid action spaces. However, these designs often suffer from low exploration efficiency and unstable training due to the infinite action spaces and interference between different sub-actions Bester et al. (2019), respectively. Action space shaping Kanervisto et al. (2020) is another way to tackle these problems. Particularly, many RL applications Kanervisto et al. (2022); Wei et al. (2022) design specific action discretization mechanisms to simplify the decision-making spaces, leading to the promising performance improvement, but it requires intensive investigations about the corresponding environments. Moreover, the combination of many manually discretized sub-actions will result in the exponential explosion of action numbers, which is incompatible with large action spaces. Recently, some works propose to learn abstract action representations to boost RL training. HyAR Li et al. (2021) designs a special training scheme with VAE Kingma & Welling (2014) to map the original hybrid action space to a continuous latent action space. Some other methods Dadashi et al. (2022); Shafiullah et al. (2022); Jiang et al. (2022) build prior sets of discrete actions to from expert demonstrations, and then deploy RL agents on this fixed discrete action sets. To preserve the necessary attributes of environments, all the above discretiza-

tion techniques require related domain knowledge to discard redundant information about actions, which means that they are unsuitable for different environments with arbitrary action spaces.

In this paper, we focus on how to learn a unified discrete action representations from scratch without any domain knowledge. Based on previous analysis and our investigations (as shown in Figure 1), we summarize the following advantages of discretization for the complexity of the action space:

- Unified action discretization provides a powerful and general approach to dealing with reinforcement learning in complex action spaces. It is equivalent to split the entire pipeline into two parts: (1). representation learning and (2). decision-making. The former focus on intrinsic properties and data distributions of the action space, then transform various action spaces into standard discrete action sets, while the latter only needs to solve core decision-making problems.

- Effective discretization can improve sample efficiency by reducing the overhead in repeating sub-optimal, useless, and semantically similar actions. RL agent can just explore and exploit the necessary subsets of the original action space during training.

Then, we introduce Neural Discrete Reinforcement Learning (NDRL) framework. Specifically, inspired by VQ-VAE van den Oord et al. (2017), we propose a action representation method called Action Discretization Variational Auto-Encoder (AD-VAE) to learn latent discrete action space from the original environment, and conduct RL on the learned space utilizing any classical RL techniques about the discrete action. It is essential to capture the intrinsic properties of the original action space, which is beneficial to learn a compact latent action space while keeping necessary information of the original action space. Therefore, we design a state-conditioned action encoder and decoder, and utilize graph neural network (Kipf & Welling, 2016) and soft-argmax operation Luvizon et al. (2019) to improve the capability of AD-VAE for the relationships between different action dimensions and boundary action values. Furthermore, we find a core issue of parallel optimization of AD-VAE and RL agents: the online updates of AD-VAE may lead to semantic changes of latent actions (i.e. the non-stationary of decision spaces), resulting in severe data staleness and Q-value over-estimation. To solve this problem, we introduce action remapping and ensemble Q-learning. Concretely, we apply the classic DQN as an instance to our framework, named **A**ction **D**iscretization **Q**-learning (**ADQ**), which can be deployed for most complex action spaces. Compared with pioneer works (Chandak et al., 2019a; Zhou et al., 2020; Dadashi et al., 2022), to our best knowledge, our proposed framework is the first online RL paradigm capable of employing in discrete action spaces learned from different continuous and hybrid decision-making environments.

To demonstrate the efficiency and stability of our NDRL framework and AD-VAE method, we evaluate it on the classic continuous control benchmark MuJoCo Todorov et al. (2012), showing that ADQ can achieve excellent performance operating in high-dimensional continuous space even with a small number of actions. To evaluate the generality, we test it on the hybrid action environments Gym Hybrid thomashirtz (2021), HardMove from HyAR and GoBigger Zhang (2021). The results show that ADQ outperforms current state-of-the-art hybrid action algorithms in both sample efficiency and final performance. Besides, we also conduct a series of ablation study experiments and interpret more details about NDRL by visualization on the latent space.

## 2 RELATED WORK

**Action Discretization** Discretization and continuity are like the relationship between 0 and 1 in the binary world. All things, including time and space, are continuous, but for the convenience of cognition, we will discretize all of them. Only then can we have measures, such as the concepts of hours, minutes and meters. In RL, learning directly on a high-dimensional continuous action space may present difficulties in exploration due to the uncountable set of actions. In addition, (Bjorck et al., 2021) argues that the nonlinear function saturation caused by unstable network parameterization will cause the well-known high variance problem. The most straightforward solution is discretization, however, this usually suffers from the curse of dimensionality. To alleviate this problem, many assumptions about the action space have been proposed. For example, (Tang & Agrawal, 2020) verifies the feasibility of discretizing the action space in on-policy optimization by utilizing the factorized distribution across action dimensions. In (Dadashi et al., 2022), the authors proposed to circumvent the curse of dimensionality problem by learning a set of plausible discrete actions from expert demonstrations. We argue that this algorithm can naturally be seen as a special case of

our NDRL framework. (Seyde et al., 2021a) explored the effect of extreme actions on continuous action control, which also inspired the design of AD-VAE for maintaining boundary actions.

**Hybrid Action Space**   Many real-world problems may have hybrid action spaces. For example, in the GoBigger game, we need to select an action type first, and then give its corresponding continuous control arguments. The simplest idea is to map it onto a unified homogeneous action space, like discretizing continuous actions or making discrete actions continuous, however this may create scalability issues with the curse of dimensionality. Going a step further, recent work proposes various hand-designed network structures to learn directly on the original hybrid action space. For instance, Parameterized Action DDPG (Hausknecht & Stone, 2016) uses a modified DDPG actor-critic structure and HPPO (Fan et al., 2019a) proposes different types of heads for different action types. PDQN (Xiong et al., 2018a) and MPDQN (Bester et al., 2019) use a hybrid structure of DQN and DDPG, explicitly modeling the dependencies between continuous and discrete sub-actions.

**Action Representation Learning**   The concept of the latent space is widely used in various elements of reinforcement learning, such as latent state and dynamics. But in action space, (Chandak et al., 2019b) proposes action representation learning in a large action space, leveraging the structure in the space of actions and showing its importance for enhancing generalization over large action sets in real-world large-scale applications. (Li et al., 2021) propose Hybrid Action Representation (HyAR) to learn a compact and decodable latent representation space for the original hybrid action space. HyAR constructs the latent space and embeds the dependence between discrete action and continuous arguments via an embedding table and conditional Variational Auto-Encoder (VAE).

## 3   BACKGROUND

**Markov Decision Process**   In reinforcement learning, we model a decision-making problem as a Markov Decision Process (MDP) $\mathcal{M}=(\mathcal{S}, \mathcal{A}, \mathcal{P}, \mathcal{R}, \gamma, \rho_0)$, where $\mathcal{S}$ and $\mathcal{A}$ represent the state space and the action space, $\mathcal{P}$ is the transition function: $\mathcal{S} \times \mathcal{A} \rightarrow \mathcal{S}$, $\mathcal{R}$ is the expected reward function: $\mathcal{S} \times \mathcal{A} \rightarrow \mathbb{R}$, $\gamma \in [0, 1)$ is the discounted factor, and $\rho_0$ is the initial state distribution. The objective of RL is to learn a policy $\pi : \mathcal{S} \rightarrow \mathcal{A}$ to maximize the expected discounted return $J(\pi) = \mathbb{E}_{\pi, \rho_0, \mathcal{P}, \mathcal{R}}[\sum_{t=0}^{\infty} \gamma^t r_t]$, where the expectation is taken with respect to the trajectory distribution induced by $\pi$ and environment dynamics.

**Hybrid Action Space**   In decision-making problems, at each time $t$, the agent receives a state and carries out an action, which can be divided into 3 types: discrete, continuous and hybrid action. Here we give a general formalization. A hybrid action $a$ contains $N$ decision nodes (sub-actions). At each decision node $i$ and time step $t$, the agent needs to give a *proto-action* $a_{t,i}$ with two attributes including type and range of values; the type of value $a_{t,i,type}$ indicates whether it is continuous or discrete, and the range of value $a_{t,i,range}$ indicates its corresponding executable action set. Thus, we use an ordered tuple like $a = (a_{t,1}, a_{t,2}, ..., a_{t,N})$ to describe these basic nodes. Furthermore, the relations between decision nodes can be defined by a adjacency matrix $A_{r,t}$ in graph theory, called action relation matrix. The value of the elements in the matrix is $\{0, 1\}$. If the element $A_{i,j}$ in row $i$ and column $j$ is equal to 1, it means that there is a directed edge from decision node $i$ to $j$. If equal to 0, there is no dependency between them. In many real-world problems, the dependencies between the *proto-action* always are invariant, that is, the action relation matrix is independent of $t$. Generally, hybrid action space can be defined as a tuple:

$$\mathcal{A} = (\{a_{t,i,type}, a_{t,i,range} \mid i \in [1, ..., N]\}, A_r) \tag{1}$$

The Parameterized Action Space defined in (Masson et al., 2016) is a special instance of our definition, specifically, which is equivalent to action $a$ containing 2 *decision nodes* and $a_{t,0,type} = 0$, $a_{t,0,range} = \mathcal{K}$, $a_{t,1,type} = 1$, $a_{t,1,range} = \mathcal{X}$. There is only a directed edge from *decision node* 1 to *decision node* 2, formally, the adjacency matrix $A_r$ is:

$$\begin{pmatrix} 0 & 1 \\ 0 & 0 \end{pmatrix} \tag{2}$$

**Action Transformed MDP**   Here we augment a MDP with action transformation, which can be defined as $\mathcal{M}=(\mathcal{S}, \mathcal{A}, \mathcal{P}, \mathcal{R}, \gamma, \rho_0, \mathcal{T})$, where $\mathcal{T}$ denotes the transformation operator on action space, such as action discretization . Denote the transformed action as $k$, we can describe $\mathcal{T}$ as:

$$\mathcal{T} : k = \mathcal{T}(s, a) \ \ a = \mathcal{T}_{-1}(s, k) \tag{3}$$

The other elements are consistent with the original definition of MDP. Through this transformation, we can learn an RL agent more efficiently on this new, often reduced, latent action space.

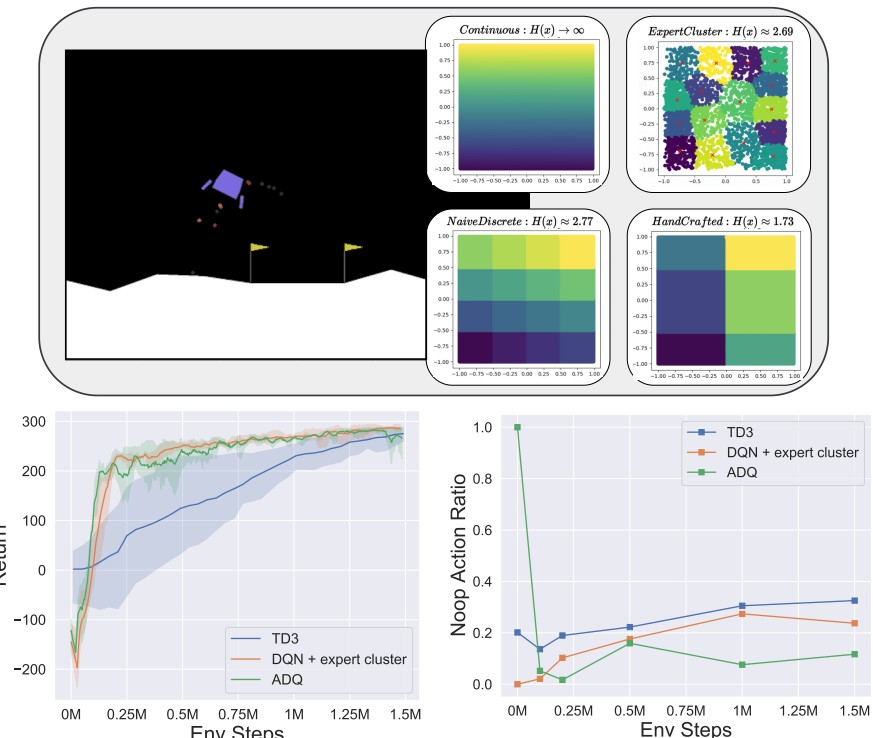

Figure 1: **Top**: Visualization analysis about different action spaces in the same LunarLander (Brockman et al., 2016) environment. *ExpertCluster* is discrete action obtained by clustering on TD3 expert data). *HandCrafted* is obtained by the threshold of spaceship engine. Vertical engine only enables when x is bigger than 0; And if y is smaller than -0.5, the left booster will fire, and if y is bigger than 0.5, the right booster will fire. **Bottom**: (left) Episode return of three algorithms on LunarLander: TD3 (original continuous action space), DQN + expert cluster, ADQ (discrete action learned by AD-VAE from scratch) ; (right) The ratio of some semantically same actions (no operation), more *no-op* actions means greater redundancy during training, i.e., lower is better.

**Vector Quantised Variational AutoEncoders**    Motivated by vector quantization (VQ) and Variational AutoEncoder, VQ-VAE van den Oord et al. (2017)) is designed to learn a discrete latent representations to represent the original data distribution (*e.g.*, text, image) in a unsupervised manner. VQ-VAE mainly comprises of an encoder $e_\phi$, an decoder $d_\psi$, and a learnable code table $V_\varepsilon$. The learnable code table maintains a set of embeddings $\{e_k\}_{k=0}^{K-1}$.

Firstly, the encoder takes the data $x$ as input, and outputs an embedding vector $z^e = f(x)$. Then using the embedding vector $z^e$ to query the nearest (usually in Euclidean distance) code vector $z^d$ in the code table and outputs an latent index $k$ simultaneously. Thirdly, the decoder uses the code vector $e^d$ as its input to produce reconstructions $\hat{x}$. The whole objective is to minimize the following loss function

$$\mathcal{L} = \mathcal{L}_d(\hat{x}, x) + \|\mathrm{sg}\,[e_\phi(x)] - z^e\|_2^2 + \beta\,\|e_\phi(x) - \mathrm{sg}[z^e]\|_2^2 \tag{4}$$

$$z^d = e_k, \text{ where } k = \mathrm{argmin}_j\,\|z^e - e_j\|_2 \tag{5}$$

Where $sg(\cdot)$ is the stop gradient function. The first term of loss function is to reconstruct error in certain distance metric, the second item and the third term is embedding loss and commitment loss respectively. Please refer to van den Oord et al. (2017) for more details.

## 4 NEURAL DISCRETE REINFORCEMENT LEARNING

### 4.1 MOTIVATION

First, to motivate our proposed framework, we further discuss the advantages of reinforcement learning in discrete action spaces from the following 3 aspects: unity, efficiency and stability.

### 4.1.1 UNITY

In practice, researchers need to first transform the original decision-making problem into a standard MDP form. Due to the different types of target action spaces, it is inevitable to utilize different techniques for the corresponding action spaces, which brings non-negligible learning and tuning costs beyond core RL optimization. But when we dive deeper into this problem, we find there are obvious redundancies in most complex action space, e.g., only a few discrete actions/samples can perform well in multi-dimensional continuous control Seyde et al. (2021b); Hubert et al. (2021). In Figure 1(a), we also illustrate the entropy of different action spaces to show the effectiveness of discretization. Therefore, learning compact action representations instead of repeating some dirty work in the raw action space is a natural and powerful choice. Moreover, the decoupling of action representation learning and RL allows researchers to concentrate on only one of the topics.

### 4.1.2 EFFICIENCY

Furthermore, we verify the efficiency of action discretization in online RL training. As shown in Figure 1(a), we find the well-designed discrete action space shows lower entropy and more compact representation. We also conduct a simple experiment to test the performance and the ratio of useless action like excessive *no operation* in 1(b). Based on these observations, we design AD-VAE to automatically learn the latent discrete action space from the original action space. On one hand, this model can learn to approximate the necessary parts and ignore meaningless parts of the original space, which is beneficial to improve exploration efficiency. On the other hand, some works Jiang et al. (2022) show that the marginal distribution of each action dimension is often multi-modal, thus using a discrete categorical distribution is more suitable than the simple regression in TD3.

### 4.1.3 STABILITY

However, previous methods Dadashi et al. (2022); Jiang et al. (2022) only succeed in deploying action discretization on imitation learning and offline RL settings, which means that it needs to first learn a latent space and then apply decision-making algorithms on the fixed discrete action space. We tried to directly utilize AD-VAE on online settings but obtain unstable episode returns. Compared to training on the frozen discrete action space, we find some abnormal indicators including unusually high Q-values and obvious fluctuations in gradient scale and variance. Due to parallel optimization about AD-VAE and RL agents, it would be a hazardous non-stationary MDP if the latent action space changes too much. To figure out this problem, we propose action remapping and ensemble Q-learning. Together with AD-VAE, these techniques form the entire NDRL framework.

## 4.2 NDRL FRAMEWORK

### 4.2.1 OVERVIEW

Motivated by the above analysis, we propose a framework that combines online RL training with learnable action discretization on complex action spaces, named **N**eural **D**iscrete **R**einforcement **L**earning (NDRL). At a high-level, this framework is a "meta-algorithm" that splits decision-making in arbitrary complex action spaces into two parts: the part of representation learning mapping the original action space to a new discrete action space and the reinforcement learning part built on learned discrete representations. Based on this design, any reinforcement learning designed for discrete space (e.g. DQN, PPO) can be potentially applied to different complex action spaces. The overview of NDRL framework is described in Figure 1, and we will introduce the data collecting phase and network training phase respectively:

**Collecting Phase**: This phase describes how to collect data used in the training of AD-VAE and RL agents. Given the current state $s_t$, RL agents first select corresponding discrete latent action $k_t$, then utilizing AD-VAE decoder to transform it back to the original action space. Moreover, NDRL also deploys some extra randomization operations (e.g. epsilon greedy in DQN) on the decoded action to maintain sufficient exploration. The final action $a_t$ interacts with the environment and it returns reward $r_t$ and next state $s_{t+1}$. All necessary data will be packed into a transition and put into buffers.

**Training Phase**: The training phase is to execute two training pipelines with different data buffers in parallel. (1) For action representation learning, NDRL follows the main training scheme of VQ-VAE to reconstruct primitive actions with state conditions. (2) For RL training, it first remaps the

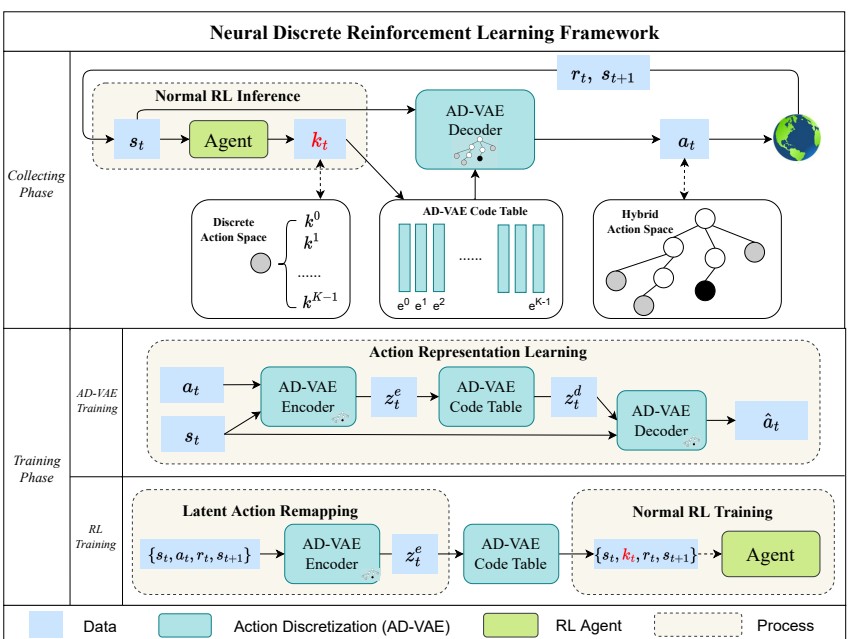

Figure 2: Overview of NDRL framework. **Collect Phase**: Given the current state $s_t$, the RL agent gives latent discrete action $k_t$, and utilizes AD-VAE code table and decoder to obtain the raw action $a_t$, then interacts with the environment. **Train Phase**: First remap the sampled transition $(s_t, a_t, r_t, s_{t+1})$ with latest AD-VAE encoder and table to $k_t$, then deploy the normal RL training. Note the AD-VAE is also trained with state condition in parallel. Black round means continuous action space, while grey round means discrete, the whole tree structure means hybrid action.

original action in the sampled transition with AD-VAE encoder to obtain the latest latent action $k_t$, then conduct normal policy optimization on transformed data.

Besides, for the better initialization of off-policy RL algorithms, we also design a pretrain stage at the beginning of the entire algorithm. The full pseudo-code of NDRL is provided in Algorithm 1. If there are some expert demonstrations, a promising set of discrete action candidates can be learned from it. Otherwise, we can collect data with random policy to train the AD-VAE and learn some basic properties of the original action space for subsequent parallel learning. After pretrain stage, data collecting and two parts of training can be executed asynchronously, so the computational cost of NDRL can be easily optimized and show the same efficiency as other methods.

### 4.2.2 AD-VAE

In this section, we first analyze the problem of directly using the original VQ-VAE, then introduce the specific design about our proposed Action Discretization Variational AutoEncoders (AD-VAE).

**Modeling Intrinsic Properties of Action Spaces** In online RL training, there are both necessary and redundant subsets of the original action space. For instance, Seyde et al. (2021b) pointed out that in some continuous action tasks, the optimal action may be at some extreme boundary values (e.g. -1 and 1), which is a common phenomenon in several physical simulation environments. Also, Bester et al. (2019) revealed that effective combinations between different action parts is significant for the optimization in hybrid action space. Therefore, it is wise to pay more attention to those actions that are more beneficial to the optimal policy, and ignore some useless even harmful actions. Otherwise, trivial action reconstruction with the same weights and distance metrics can only obtain some over-smooth actions. Besides, we also can take advantages of the value function to focus on those actions with higher future return, saving the cost of agent exploration and exploitation.

**Information Completion in AD-VAE** We first introduce the technique of information completion in AD-VAE. In some complex environments, the set of optimal actions could be large and vary in different training stages. If we reconstruct actions without state information, AD-VAE must maintain a large discrete action sets and RL agents needs to learn decisions on a great number of actions, which can easily make it overwhelming and cause training instability. On the contrary, properly use of state-conditioned input in both encoder and decoder of AD-VAE can increase the representational

power and diversity, and reduce the burden of action reconstruction and RL training. Moreover, the relationship between different action parts, i.e. the action relation matrix mentioned in Section 3, can assist AD-VAE to learn more efficiently and reasonably, so we can represent this information as the connections of nodes with a graph neural network. Besides, AD-VAE is not designed to learn the entire action space but to properly model the subsets required by current RL optimization, thus we customize sampling method and training scheme for AD-VAE, including sampling a mixture of stale data in replay buffer and latest collected data as a training mini-batch, validating the reconstruction error of actions to indicate the update frequency.

**Continuous Action Regression in AD-VAE** Another important intrinsic property of original action spaces is some special action values, such as the extreme actions mentioned in Seyde et al. (2021b) or thresholds of engine dynamics in LunarLander. It is critical for action representation networks to restore these continuous value accurately. Therefore, in AD-VAE, we adopt the soft-argmax operation to automatically reconstruct these special actions. Assuming the range of original action is $[A_{\min}, A_{\max}]$, and it is divided into $N + 1$ bins on average, the predicted action is:

$$\hat{a} = \sum_{j=0}^{N} s_j * p_j(s_i), s_j = A_{\min} + j * (A_{\max} - A_{\min})/N \tag{6}$$

When evaluation, we directly output the corresponding support value if the probability of the support is greater than a threshold (e.g. 0.9). In some environments like *Hopper/Halfcheetah*, we find this design can help a simple DQN agent achieve a comparable performance with TD3.

Other parts of AD-VAE follow the design of VQ-VAE, the whole training procedure is to minimize the loss function described in Equation 4.

### 4.2.3 ADAPTING RL TO LATENT ACTION SPACES

In this section, we continue to analyze why and how to adapt online RL to latent discrete action spaces, and then illustrate an instance of our NDRL framework on the value-based RL algorithm DQN, Action Discretization Q-learning (ADQ).

**Semantic Inconsistency** In online RL, Double DQN (van Hasselt et al., 2016) pointed out that Q-value over-estimation problems caused by the function approximation error and the max operator in the bootstrap target often lead to performance deterioration. Furthermore, this problem may be exacerbated in NDRL. On one hand, latent action stored in replay buffer will be stale and biased due to updates of AD-VAE. On the other hand, since AD-VAE is dynamically and simultaneously updated together with RL agents, for a particular latent action, the corresponding action in the original action space could often change, which is more likely to lead to the overestimation of Q-value.

**Latent Action Remapping** Similar to the reanalyze operation in MuZero Schrittwieser et al. (2019), we design a latent action remapping operation to solve the problem of stale data. In the collected mini-batch $\{s_t, a_t, k_t^{old}, r_t, s_{t+1}\}$, the latent action is determined by the old version of AD-VAE. When updating RL agents, we remap the original action to the corresponding latent action via the latest action encoder $e_\phi$: $k_t = e_\psi(s_t, a_t)$, and then executes RL training on the remapped samples $\{s_t, sg[k_t^{new}], r_t, s_{t+1}\}$ ($sg$ means the stop gradient operation).

**Ensemble Q-learning** To further alleviate the *Semantic Inconsistency* problem, inspired by previous work Anschel et al. (2017); An et al. (2021), we propose an ensemble Q-learning method for more stable Q-value updates for latent action space, reducing the uncertainty of approximation error and over-estimation, which greatly improves the stability of parallel optimization of AD-VAE and RL agents. Specifically, we utilize a shared state encoder and N ensemble Q-value heads, i.e., the penultimate layer of the Q network is connected to N linear layers and outputs N Q-value, then we adjust the update equation of Double DQN as follows:

$$L_i = \left[Q(s_t, a_t; \theta_k) - [r_t + \gamma min_k Q(s_{t+1}, a_{t+1}; \hat{\theta}_k)]\right]^2 \tag{7}$$

$$a_{t+1} = argmax \frac{1}{N} \sum_k Q(s_{t+1}, a_{t+1}; \theta_k) \tag{8}$$

Where $L_i$ means the loss function of *i-th* Q head. Detailed experiments can be found in Section 5.3.

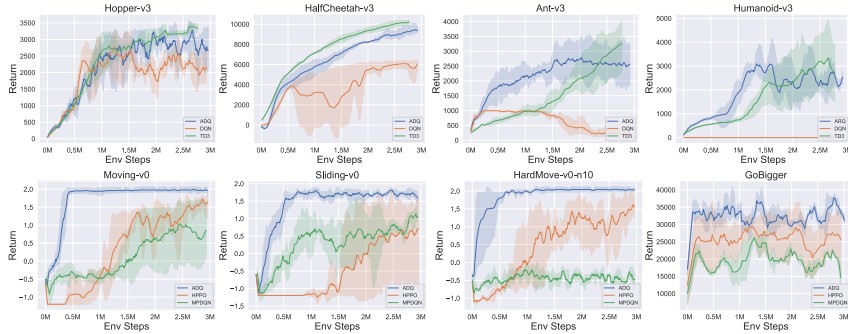

Figure 3: Training curves of ADQ against other baseline algorithms in environments with complex action spaces. **Top**: In four continuous action environments of MuJoCo, ADQ significantly outperforms DQN with manually discretized action space, and is basically comparable to the classic TD3 algorithm. **Bottom**: In four hybrid action environments in Gym-Hybrid, HardMove and GoBigger, ADQ outperforms the baseline MPDQN and HPPO in both performance and stability. Curves and shadings denote the mean and standard deviation over 5 seeds.

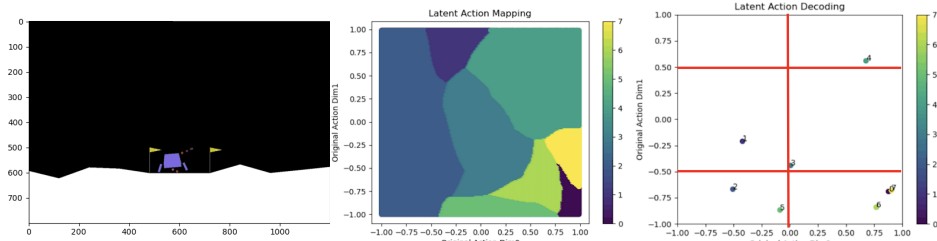

Figure 4: Visualization of the latent action space of LunarLander games. Details is in 5.2.

## 5 EXPERIMENTS

For the evaluation of our NDRL framework, we ask and answer the following questions: 1) Is it efficient and stable to employ online RL training on discretization action spaces for different decision problems, especially in high-dimensional continuous and hybrid action spaces? (Section 5.1); 2) How do we interpret the training of the latent action space? Can we further verify some observations mentioned in introduction parts? (Section 5.2); 3) How do various designs improve the NDRL framework, including AD-VAE and other RL adaption techniques? (Section 5.3).

### 5.1 MAIN RESULTS

In this section, we investigate the performance and efficiency of NDRL in various continuous and hybrid action environments against previous algorithms designed specifically for these action spaces. Firstly, we evaluate our methods on MuJoCo, a classic continuous control benchmark, including two high-dimensional continuous domains (Ant and Humanoid with 8 and 17 dimensions respectively). Note we also add a few redundant dimensions in original action spaces. We set up two comparison groups for our ADQ, one is the popular continuous action space algorithm TD3, and the other is naive DQN deployed in the manually discretized action space, i.e., equally dividing the original continuous action into 3 bins at each dimension and using their Cartesian product to obtain handcrafted discrete actions. In Figure 3, ADQ can acquire comparable results to TD3 and show a obvious improvement over naive DQN in all four domains on the top. Secondly, we leverage Gym-Hybrid, HardMove and GoBigger to verify the effectiveness of ADQ in more complex hybrid action spaces. This requires to deal with the relationship between different actions parts (e.g., the value of action arguments depend on the choice of action type) and more complex environment dynamics (GoBigger). At the bottom of Figure 3, we compare ADQ with two types of hybrid action space algorithms, MPDQN and HPPO, suggesting that ADQ can automatically learn the intrinsic properties of discrete action type and continuous action arguments, and performs excellent performance and solid stability. All the detailed settings are shown in Appendix **??** and **??** respectively.

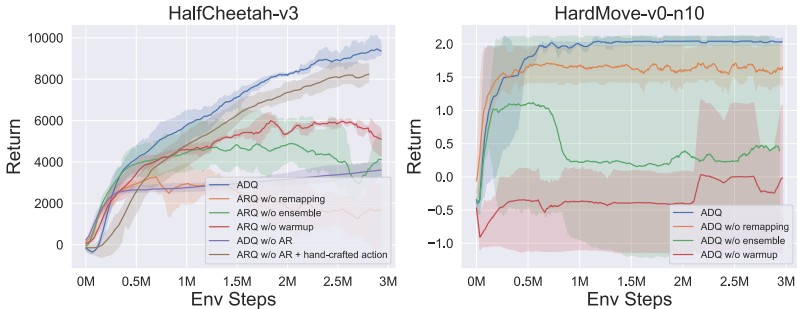

Figure 5: Ablating results of ADQ in two types of environments (continuous and hybrid) over 5 seeds. When we remove any one of the proposed techniques, the performance of ADQ will drop significantly, which verifies the effectiveness of our proposed techniques. *ADQ w/o AR* is especially important for improving ADQ performance in environments such as *HalfCheetah-v3*.

## 5.2 VISUALIZATION ANALYSIS OF LATENT ACTION SPACE

Furthermore, we demonstrate the learned latent action representation in LunarLander environment for 2-dimensional continuous control. Figure 4 first shows the status of the spaceship when it is about to land (left), i.e., launching both horizontal and vertical engines to control speed and position, then uniformly samples points in original continuous space and transforms them with AD-VAE encoder to find their nearest discrete indexes (middle). Also, we directly send the corresponding embeddings in code table to decoder to acquire their counterparts in the raw space (right). We can observe that the learned latent space is similar to the intrinsic mechanisms of this environment, highlighted by red line: only using one discrete action to represent less important actions in this state like *no-op*, mapping several different actions to the bottom right corner.

## 5.3 ABLATION STUDIES

We also empirically evaluate the specific impacts of our proposed AD-VAE and RL adaption techniques on two example environments respectively: *HalfCheetah-v3* (continuous), and *HardMove-v0-n10* (hybrid). The ablation results are shown in Figure 5. Concretely, we have the following five ablation variants in total, and their brief descriptions are as follows:

**ADQ w/o remapping**: ADQ variant that does not remap latent actions during RL training.

**ADQ w/o ensemble**: A variant of ADQ that doesn't utilizing the *Ensemble Q-Learning* technique.

**ADQ w/o warmup**: ADQ variant that starts training without any warmup pretraining.

**ADQ w/o AR**: ADQ variant with traditional VQ-VAE reconstruction head.

**ADQ w/o AR + hand-crafted action**: Built on **ADQ w/o AR**, this variant adds manually selected boundary actions (i.e. the Bernoulli extreme actions) to latent discrete action spaces for RL training.

Figure 5 show that when we remove either of the proposed techniques, the performance of ADQ drops significantly in both two environments, verifying the effectiveness of our proposed techniques. Due to the semantic inconsistency problem, the **ADQ w/o ensemble** agent suffers from severe over-estimation issues and finally show poor performance. The **ADQ w/o remapping** agent meets the same problem, but in some cases the over-estimation problem can be partially alleviated by *Ensemble Q-learning* technique. The **ADQ w/o warmup** agent shows much slow learning progress due to lack of good starting points. Note that in environments with boundary optimal actions such as *HalfCheetah-v3*, **ADQ w/o AR** is especially important for improving ADQ performance, even better than the variant using extra hand-crafted actions. Other ablation results like pretraining on expert demonstrations and the sensity of hyper-parameters can also be found in Appendix.

## 6 CONCLUSIONS AND LIMITATIONS

Starting from comprehensive analysis for action discretization, we introduce a general and efficient paradigm named Neural Discrete Reinforcement Learning, including our proposed AD-VAE and RL adaption techniques. We empirically evaluate the efficiency and stability of our framework. Although our method achieve superior performance in different benchmark environments, there are still some challenging action spaces in multi-agent games, such as variable-length actions in episodes. Besides, combining latent discrete actions with MCTS is also a valuable attempt. We will continue to pursue ultimate solution for action space shaping in future work.

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
