# OpenReview forum: "Neural Discrete Reinforcement Learning"
_ICLR.cc/2023/Conference — Submitted to ICLR 2023_

### Official Review · Reviewer_zzhQ · 2022-10-21

**Confidence:** 3
**Correctness:** 3
**Technical Novelty And Significance:** 3
**Empirical Novelty And Significance:** 3
**Recommendation:** 5

**Clarity, Quality, Novelty And Reproducibility:**

Detailed review:

The paper is somewhat clear, and the proposed method seems novel. However, in its current form, the quality of the method is hard to justify due to the missing details.

How is the TD3 ExpertCluster obtained (Figure 1)? It needs to be clarified why TD3 is chosen as the algorithm for continuous action space. In the context of DQN, DDPG seems a reasonable choice to show how the task can be solved using the continuous version.

It is unclear to me what is the main contribution of the paper. Is it a framework (NDRL) or a method AD-VAE for action representation (discretization?), or a better algorithm (ADQ) for the continuous high-dimensional task?

If it’s a framework, then at least two algorithms need to be demonstrated to show that the framework is effective. As it is now, I can see DQN-based algorithm is incorporated.

For the action representation or discretization, an equivalent algorithm should be compared. For example, the proposed ADQ is compared with discrete-action DQN and continuous-action TD3. However, it is unclear if the action discretization is helpful from the comparison of ADQ vs. TD3. A reasonable comparison would use DDPG, a continuous action version of the DQN.

A suggestion to the author(s) is to use PPO in evaluating the action discretization. PPO can be applied to discrete and continuous action spaces with minimal algorithmic changes.
In performance, the action representation should help in getting better performance compared to continuous action. However, if the performance remains the same, then that conflicts with the motivation of the paper, that is:
“However, the complexity of action spaces still prevents us from directly utilizing advanced RL algorithms to real-world scenarios, such as high-dimensional continuous control in robot manipulation.”
Why do we still need discretization if we have a better continuous-action algorithm?


I am a bit confused with the use of Discrete Action Space (k^0, ..,k^{K-1}) in Figure 2. My understanding is that this is used for the remapping step in RL training. So how do you get this Discrete action space for a continuous control task?
Algorithm 1 is mentioned, but the details discussion is missing. How are the k_t and a_t calculated in Algorithm 1 (Appendix) lines 18 and 20?


**Strength And Weaknesses:**

Strength:
The problem of action discretization seems interesting as it allows the reuse of effective discrete action-based RL algorithms to continuous action tasks.
The choice of evaluation on high-dimension continuous tasks seems interesting.

Weakness (see details comments):
Important algorithm details need to be included, which hampers the evaluation of the paper's contribution.
Lack of evaluation to justify the effectiveness of the proposed method.
Overall, the paper needs more coherency in writing. In many places, the motivation and the methodology are tangled.


**Summary Of The Paper:**

This paper proposes an action representation learning method ( AD-VAE)  to learn compact latent action spaces to discretize the action spaces for reinforcement learning training. Furthermore, the paper proposes a few techniques (latent action remapping, ensemble) to mitigate the instability of AD-VAE while training with RL algorithms. The method is demonstrated with a Q-learning-based algorithm. The methods are evaluated on several continuous and hybrid action-space reinforcement learning tasks.

**Summary Of The Review:**

Important algorithm details are missing, which hampers the evaluation of the paper's contribution. In addition, the evaluation lacks important experiments that can justify the proposed methods' effectiveness. Overall, the paper needs more coherency in writing. In many places, the motivation and the methodology are tangled, eventually hampering understanding of the methodological contribution.

---

> ### Author Response · Authors · 2022-11-19
> **Response to Reviewer zzhQ**
>
> Thank you very much for your feedback and suggestions, we will address your concerns and questions as follows:
>
> > Question about the main contribution:
>
> Our main contribution is the proposed unified NDRL framework to address decision-making problems in various complex action spaces, including hybrid action spaces and high-dimensional continuous action spaces.
> Specifically, we propose AD-VAE as our action representation learner, discover and analyze some problems in online joint action representation and reinforcement learning, such as data staleness and Q-value overestimation caused by semantic changes of latent actions. The proposed feasible algorithm instance ADQ fuses multiple practical techniques, achieving new SOTA performance on hybrid action spaces, comparable to the best continuous action space algorithm TD3.
>
> > Question about writing of  Figure 1 and Algorithm 1:
>
> Figure 1: We train a TD3 agent on the lunarlander (continuous action) environment, then use the TD3 agent to collect 200 episodes on 20 seeds in total, next we cluster the actions in this collected dataset with k-mens algorithm (K=16), obtain 16 continuous actions, we called them TD3 ExpertCluster actions. Then we map the (K=16) discrete actions of the DQN algorithm to these 16 fixed continuous actions, thus we can train a DQN agent on this lunarlander (continuous action) environment. Empirically, we found that the learning curve of this DQN agent is the same as our proposed ADQ algorithm, which verifies that the ADQ agent can automatically learn a similar set of expert actions as the manually obtained TD3 ExpertCluster atcion set.
> Algorithm 1: The discrete action space is learned by AD-VAE, that is, the mapping relationship between the original action space and the discrete action space is learned. k_t is the discrete action, which is given by any discrete action space reinforcement learning algorithm such as DQN, and a_t is the original action decoded by k_t through the learned AD-VAE decoder. It should be emphasized that our AD-VAE is applicable to arbitrarily complex action spaces, including hybrid actions and high-dimensional continuous actions.
> The details of Figure 1 and Algorithm 1 are further explained and analyzed in the revised paper.
>
> > Question about the choice of baseline algorithm for comparison:
>
> Continuous Action Space: TD3 [1] is an improved version of DDPG [2], and TD3 performs much better than the original DDPG in mujoco environment. So we choose TD3 as our baseline comparison algorithm. Our ADQ achieves comparable results to TD3 on continuous-action environments and clearly outperforms DDPG. Furthermore, we also provide here [https://anonymous.4open.science/r/NDRL-benchmark-01E5/] additional baseline comparison results with other action space discretization methods, such as [3] [4]. We can see that our proposed ADQ outperforms other baseline algorithms in most environments.
> Hybrid Action Space: We choose the current SOTA algorithms MPDQN [5] and HPPO [6] as our baseline comparison algorithms, and we outperform them both in convergence performance and stability in gym-hybrid and GoBigger environments.
>
> > Question about different algorithm instances:
>
> As a framework aimed at solving arbitrarily complex action space problems, NDRL has the ability to combine different representation learning and reinforcement learning algorithms. In this paper, we choose off-policy DQN as our baseline RL algorithm.
> Due to the need to use off-policy data and joint online updates, in principle,  combined with off-policy DQN algorithm will be more difficult than combined with on-policy PPO algorithm, e.g.  the online updates of AD-VAE may lead to semantic changes of latent actions (i.e. the non-stationary of decision spaces), resulting in severe data staleness and Q-value over-estimation.
> Currently, combined with many proposed stabilization and boosting techniques, the DQN-based algorithm instance ADQ has achieved good performance in continuous and hybrid action space environments.
> Therefore, we guess that it should be relatively easy to combine the on-policy PPO algorithm in the NDRL framework. However, due to time constraints, we will submit more additional experiments on NDRL+on-policy PPO in a revised paper by the end of November.
>
> [1] https://arxiv.org/pdf/1802.09477
>
> [2] https://arxiv.org/abs/1509.02971
>
> [3] Tavakoli et al. (2018) Action branching architectures for deep reinforcement learning. AAAI.
>
> [4] Tavakoli et al. (2021) Learning to Represent Action Values as a Hypergraph on the Action Vertices. ICLR.
>
> [5] https://arxiv.org/abs/1905.04388
>
> [6] https://arxiv.org/abs/1903.01344

---

### Official Review · Reviewer_qLKy · 2022-10-24

**Confidence:** 4
**Correctness:** 2
**Technical Novelty And Significance:** 3
**Empirical Novelty And Significance:** 3
**Recommendation:** 3

**Clarity, Quality, Novelty And Reproducibility:**

- The approach is novel and in my view quite justifiable. However, I think the experiments leave a lot to be desired. I think that the experiments do not necessarily depict an advantage for their adaptive discretization method over a naive discretization scheme (see my reasons in the above section).

- There are missing references that would help depict a better picture of what subproblems the method is addressing. For instance, the problem of learning about synergies between different action dimensions has been studied by Tavakoli et al. (2021) [4], but the paper does not discuss this problem in detail and how not capturing such relationships could bias performance. I know that the authors refer to it but, in my opinion, to a reader unfamiliar with such problems it will not fully be clear exactly what the authors are referring to.

- Relationship/connection to *Action Hypergraph Networks* [4] is not discussed; e.g., "Therefore, we design a state-conditioned action encoder and decoder, and utilize graph neural network (Kipf & Welling, 2016) and soft-argmax operation Luvizon et al. (2019) to improve the capability of AD-VAE for the relationships between different action dimensions and boundary action values." Here, what is the role of GNNs in capturing such relationships? Isn't that similar to that captured by *action hypergraph networks*? And if they are related, why a graph topology should be preferred over a hypergraph formulation? A graph captures pairwise interactions/relationships, but what if a relationship exists on a higher-order combination (where a hypergraph could capture)?

- What do you mean by hybrid action spaces? I often see this term used to refer to action spaces that have discrete and continuous components.
Statements like this are unclear to me: "*Xiong et al. (2018b); Fan et al. (2019b) propose some techniques to extract the relationship between different action dimensions, which is important in hybrid action spaces.*" This statement depends on your definition of hybrid action spaces. But based on what I understand of the term, this statement is invalid: the relationships are important regardless of whether the action space is hybrid or not.

- Branching DQN (BDQN) [2] reaches >3000 performance on Humanoid-v3 in 3M environment steps using only naive uniform-interval discretization, outperforming the reported performance for the proposed NDRL method with a learned discretization (refer to results here: [link](https://github.com/atavakol/action-hypergraph-networks/blob/main/data/images/physical_results.png)). Also, in Ant-v3, HGQN-r1 outperforms the proposed approach with a naive discretization, and BDQN is somewhat on par with the proposed approach again using the naive discretization.
Not comparing to such agents has in my view led to a misinterpretation of the results of the Neural Discretization method.
Note that the aforementioned methods of BDQN and HGQN-r1 are super basic additions to DQN, without any need for such complex additions!

- I couldn't find any experiments for understanding the value of the size of latent action space (size of $K$) on performance. This is quite important in my view. On this line, seeing similar plots as that of Fig 4 (middle) but for a varying $K$ would be interesting.

- **Question:** Can there be a comparison between the optimal Q-function of LunarLander and the learned latent discrete-action partitions? This could help us understand: (1) How close to an optimal discretization are we getting based on what is our important objective (to learn the $Q^*$); (2) What would be the difference in learned discretizations if the system didn't capture inter-dimensional relationships?; (3) What aspects of the inter-dimensional relationships is the system potentially missing right now?

- **Question:** If the AD-VAE training occurs before RL training, wouldn't this make RL training experience transitions invalid (as the underlying action that generated the experience would be different from the believed action by the agent)?


**Minor:**
You have used `citet` instead of `citep`. This really affects readability. E.g., "*[...] such as high-dimensional continuous control in robot manipulation Lillicrap et al. (2016) and [...]*". Here you need to use `citep` so you get *(Lillicrap et al., 2016)*.


**Strength And Weaknesses:**

**Strengths:**

- The target problem is very important: To have a unified approach for dealing with continuous-action problems by adaptively discretizing them and applying discrete-action RL methods and importantly in an online manner (not from offline or expert data).

- The approach is reasonable but requires (somewhat justifiably) a few patches (e.g. ensemble learning to deal with overestimation over a non-stationary action space)


**Weaknesses:**

- The paper is somewhat poorly written with an unclear flow at times and missing a good positioning/elaboration by connecting to many related works to better.

- The size of the latent action spaces is limited to K, which is a priori determined. This is not a major issue but nonetheless could introduce statistical bias in performance. In fact, this bias could be potentially stronger than naive discretization (equidistance discretization). In naive discretization, the number of subactions per action dimension is fixed (say 5 subactions per dimension). Therefore, the size of the effective action space ends up increasing with increasing action dimensionality (i.e. size would be $5^N$ where $N$ is # of action dimensions). But the NDRL framework assumes a fixed effective action space size with $K$ latent actions, independent of the number of action dimensions. This can be manually adjusted in principle, but the underlying RL method should be equipped with a capacity to scale with action dimensionality as well. For instance, standard DQN doesn't have such a capacity, but Sequential DQN [1], Branching DQN [2], Amortized Q-Learning [3], and HGQN-r1 (DQN + action hypergraph networks) [4] have such capacity.

- Comparison with a representative subset of scalable algorithms (such as HGQN-r1) using the naive equidistant discretization scheme would be critical to support the argument that naive discretization is not performant. Comparison with DQN (which doesn't have a generalization capacity over combinatorial action spaces) is not useful in determining whether adaptive discretization is critical over naive discretization.

- Comparison with a bang-bang or bang-off-bang discretization is needed to ensure the domains in experiments indeed require more complex learned action spaces.

- Analysis of what those learned latent actions correspond to would be needed. E.g., similar to that in Fig 7 of Tavakoli et al. (2021), can you show that in Walker2D, the latent actions are able to capture a relationship between the two hip joints?

**References:**

[1] Metz et al. (2017) *Discrete sequential prediction of continuous actions for deep reinforcement learning*. arXiv.

[2] Tavakoli et al. (2018) *Action branching architectures for deep reinforcement learning*. AAAI.

[3] Van de Wiele (2020) *Q-learning in enormous action spaces via amortized approximate maximization*. arXiv.

[4] Tavakoli et al. (2021) *Learning to Represent Action Values as a Hypergraph on the Action Vertices*. ICLR.

**Summary Of The Paper:**

The paper proposes a framework for the adaptive discretization of continuous-action spaces in an online manner, by employing a two-phase learning structure: (1) latent action space learning; (2) RL on the learned action space.

**Summary Of The Review:**

See my comments above.

Overall, the idea is quite appealing. But to see its benefits, we'd need more experiments.

---

> ### Author Response · Authors · 2022-11-19
> **Response to Reviewer qLKy**
>
> Thank you very much for your feedback and suggestions, we will address your concerns and questions as follows:
>
> > Question about the choice of baseline algorithm for comparison:
>
> It is worth noting that our experiments are carried out on MuJoCo v3 version, while the BDQ paper uses the MuJoCo v1 version, and the HGQN paper uses the PyBullet version, the definition of different versions of the environment is different, can not Direct comparison. So we reproduced the results of the paper[1] paper[2] and bang-bang bang-off-bang on the MuJoCo v3 version, and provided the results here [https://anonymous.4open.science/r/NDRL-benchmark-01E5/] , you can see that in most environments, our ADQ algorithm outperforms the above baselines both in terms of convergence speed and final performance. In addition, the methods like bang-bang bang-off-bang, as the original action space dimension grows, the discrete action space dimension grows exponentially, and the performance drops sharply. Our ADQ algorithm cleverly circumvents this problem by utilizing the learned discrete action space.
> It should be emphasized that the compared baseline algorithms can only be used in the continuous action space environment, while our algorithm example ADQ utilizes the joint optimization of the action representation learner AD-VAE and the reinforcement learning algorithm DQN, and can also solve the mixed action space task, Such as gym-hybrid and GoBigger, and achieved new sota performance on gym-hybrid, surpassing the current strong baselines MPDQN [1] and HPPO [2].
>
> > Question: If the AD-VAE training occurs before RL training, wouldn't this make RL training experience transitions invalid (as the underlying action that generated the experience would be different from the believed action by the agent)?
>
> Indeed, due to the joint training of action representation learning and reinforcement learning, the original action corresponding to the latent action is always changing, which we call the Semantic Inconsistency issue in the paper. In order to alleviate this problem, we propose Latent Action Remapping technique, that is, in the collected mini-batch, the latent action is determined by the old version of AD-VAE. When updating RL agents, we remap the original action to the corresponding latent action via the latest action encoder, and then executes RL training on the remapped samples.
> To further improve the stability of joint training, we propose the Ensemble Q-learning technique. For the experiments here, please refer to the ablation study section in the paper. Of course, there may be other more ingenious solutions waiting to be discovered.
>
> > Question about the relationships between different action dimensions:
> In theory, our algorithm is applicable to arbitrary action spaces, including hybrid action spaces and high-dimensional continuous action spaces. In the mixed action space, the input of AD-VAE includes the continuous part and the discrete part of the mixed action, which are encoded into a latent action by the AD-VAE encoder, and then the latent action can be decoded back by the AD-VAE decoder. The original mixed action, since ADQ has achieved the performance of SOTA in the mixed action space environment, we believe that this latent action has implicitly modeled the relationship between the continuous part and the discrete part of the original action. Details here Experimental analysis, which we will present in the revised version of the paper.
>
> > Question about the effect of latent action shape and the measure of optimal discretization:
>
> Indeed, the size of the action space K is an important hyperparameter, which determines the output dimension of the Q network of the reinforcement learning agent, that is, the complexity of learning and exploration. When K is too large, the learning load is obviously too large, and when K is too small, the learned discrete actions may not be enough to cover the optimal action set of the original action space. We empirically analyze the effect of K in the appendix of the paper.
> A feasible solution is to gradually increase the number of discrete bins of actions in each dimension by performing uniform manual discretization of continuous actions in each dimension, and then learn a DQN agent on it, when the performance of DQN reaches a near-optimal level. , the dimension of the discrete action set at this time can be used as a maximum value of our K, and then use the half-half method to gradually find a minimum K that achieves optimal performance.
> How to quantitatively measure the properties  of the current discretized action space is an interesting question? It may be a good idea to compare the learned Q function on discrete action space with the optimal Q function on continuous action space. We are conducting experiments and will present detailed experimental analysis in subsequent revisions of the paper.
>
> [1] https://arxiv.org/abs/1905.04388
>
> [2] https://arxiv.org/abs/1903.01344

---

> > ### Comment · Reviewer_qLKy · 2022-11-22
> > **Results do not accord with published results, questions remain unaddressed**
> >
> > Thanks for running further experiments and for your response. Unfortunately, the response does not address my questions/concerns appropriately. The additional results provided do not come with any explanation of the training setting, architecture, hyperparameters, or hyperparameter tuning procedures. I also think the authors dismissed the fact that I shared a link with the performance of the baselines I referred to on the MuJoCo-Gym-v3's Humanoid task. Based on the result I shared, BDQ reaches a performance of ~3000 which is on par or better than reported ADQ results. This signifies that the results provided cannot be trusted, and I would now further question the validity of the other results in the paper.
> >
> > Note also that a better approach to rebuttal would be to incorporate the requested additions in the paper so that you can provide details on the exact algorithmic choices made, any public implementation used, etc. regarding the additional results.

---

### Official Review · Reviewer_qGjk · 2022-10-24

**Confidence:** 4
**Correctness:** 2
**Technical Novelty And Significance:** 2
**Empirical Novelty And Significance:** 3
**Recommendation:** 3

**Clarity, Quality, Novelty And Reproducibility:**

As I discussed before, the paper writing is clear in the beginning, but the quality decreases quickly in the body of the paper. The method is novel as far as I know, the motivation for using the method is not clear, and the paper also fails to compare to reasonable baselines.

**Strength And Weaknesses:**

Strength

The proposed idea is intuitive and novel as far as I know,

Weakness

The writing at the beginning of the paper is quite clear, but the quality of writing deteriorates quickly as the paper progresses. I list a few examples below:

- what is the takeaways for Figure 1? The figure seems hastily prepared, and the caption does not fully explain what the readers should take away from the figure. Section 4.1.2 says that Figure 1 shows that well-designed action space leads to smaller entropy, but afaik, figure 1 does not discuss this at all.

- the writing in 4.1.1 is too informal. What is dirty work in raw action space?

- figure 3 is too small and I can not read the legend at all.

Also, if the proposed method is equivalent to TD3, then why do we need to use the method at all, since TD3 is relatively simple?

The paper also misses comparison to obvious baselines, such as https://arxiv.org/abs/1705.05035

**Summary Of The Paper:**

The paper proposes using a VQ-VAE to map continuous actions to a discrete latent space. The paper then applies RL training on top of the discrete latent action space.

**Summary Of The Review:**

I do not recommend acceptance due to the low quality of the writing and presentation (for example, as explained by my comments regarding figure 1).

---

> ### Author Response · Authors · 2022-11-19
> **Response to Reviewer qGjk**
>
> Thank you very much for your feedback and suggestions, we will address your concerns and questions as follows:
>
> > Question about the advantage comparing with TD3:
>
> Our main contribution is the proposed unified NDRL framework to address decision-making problems in various complex action spaces, including hybrid action spaces and high-dimensional continuous action spaces. Specifically,  the proposed algorithm instance ADQ integrates multiple practical techniques to achieve new SOTA performance on hybrid action spaces, and is comparable to the continuous action space algorithm TD3.
> It should be emphasized that the original TD3 algorithm can only be used in continuous action space environments, our algorithm utilizes the joint optimization of the action representation learner AD-VAE and the reinforcement learning algorithm DQN, thus can also solve hybrid action space tasks, such as gym-hybrid and GoBigger.
> Moreover, on high-dimensional continuous action space environments, such as Ant-v3 and Humanoid-v3, our ADQ algorithm can learn faster early in training.
>
> > Question about the other baseline algorithms:
>
> Continuous Action Space: We provide here [https://anonymous.4open.science/r/NDRL-benchmark-01E5/] the additional baseline comparison results with other action space discretization methods, such as [1] [2]. We can see that our proposed ADQ outperforms other baseline algorithms in most environments.
> Hybrid Action Space: We choose the current SOTA algorithms MPDQN and HPPO as our baseline comparison algorithms, and we outperform them both in convergence performance and stability in gym-hybrid and GoBigger environments.
>
> > Question about the writing:
>
> Thank you for your valuable advice on writing, we have polished it up in a revised version of the paper.
> Figure 1 serves as an illustrative example of our idea of discretizing the original action space:
> In the upper part of Figure 1, in order to explain that different action spaces have different entropy, we can regard the entropy of the action space as an indicator to measure the complexity of the action space. The smaller the entropy of the action space, the less overhead the reinforcement learning agent needs to learn and explore. The entropy of the hand-designed discretized action space is often small, while the entropy of the original continuous action space is large, indicating that the agent may have higher learning efficiency in the discretized action space.
> In the lower part of Figure 1, the ADQ algorithm on the learned discrete action space can achieve similar performance to the manually designed discrete action space algorithm, and both converge faster than the TD3 algorithm learned on the original continuous action space. Some potential advantages of action space discretization are analyzed to inspire our subsequent framework and algorithm design.
>
> [1] Tavakoli et al. (2018) Action branching architectures for deep reinforcement learning. AAAI.
>
> [2] Tavakoli et al. (2021) Learning to Represent Action Values as a Hypergraph on the Action Vertices. ICLR.

---

### Official Review · Reviewer_f3BE · 2022-10-24

**Confidence:** 3
**Correctness:** 3
**Technical Novelty And Significance:** 3
**Empirical Novelty And Significance:** 3
**Recommendation:** 5

**Clarity, Quality, Novelty And Reproducibility:**

While the authors’ ideas make sense at a high level, I struggled to follow the details of their proposed algorithms (see specific clarity issues above).  This negatively affected the quality and reproducibility of the paper in my view.


**Strength And Weaknesses:**

**Edits:**
- introduction, “prior sets of discrete actions to from expert demonstrations”: grammar issue?
- introduction: “utilize graph neural network” should be “utilize a graph neural network”
- introduction: “and soft-argmax operation” should be “and a soft-argmax operation”
- “Generally, hybrid action space can be defined as a tuple:” add “a” before a “hybrid”
- “Then using the embedding vector z^e…”: grammar, what noun is doing the “using”?
- “The full pseudo-code of NDRL is provided in Algorithm 1”: this is in the supplementary material, so the reference to the pseudo-code should say “in supplementary material Section …”, to avoid making the reader look futilely around the main body of the paper for the algorithm.
- bottom of page 8: “All the detailed settings are shown in Appendix ?? and ?? respectively.”

**Strengths:**
- The ablation studies are nicely done and convincing.
- The authors appear to have chosen their hyperparameters in a principled manner: based on prior work, with little (potentially-unfair-to-baselines) tuning.  When they did tune, they explored those hyperparameters thoroughly, which further strengthens this work (Section A.5).

**Small clarity weaknesses:**
- Section 3, Hybrid Action Space, first ~5 sentences (through “to describe these basic nodes.”): This is a little hard to follow.  Upon several rereads of this part, I think I fully understand, but a more rigorous approach to defining this notation would make this part of the paper clearer and easier to read.
- psi, phi, and several other symbols were not defined.  Don’t make the reader guess or infer these definitions.

**Larger weaknesses (clarity and other):**
- Section 3, Vector Quantised Variational AutoEncoders: I found this difficult to read.  See the question below and the relevant edit above for ideas to make this part clearer.
- ”In Figure 1(a), we also illustrate the entropy of different action spaces to show the effectiveness of discretization”: A minor issue is there is no 1a subfigure.  A more significant issue: is the entropy being shown, as indicated in the quote above?  How so?  Is H entropy?  Was this defined?  What is x in this figure?  It also seems to be undefined.
- On a similar note, Figure 1 and its caption are very confusing.  In addition to the issues mentioned above:
    - x and y are undefined. (The x here does not seem to be the same as the also-undefined x which appears in the top part of the figure)
    - The sentences “HandCrafted is obtained…right booster will fire” do not make sense to me.  (These sentences seem to be describing a hand-crafted policy, but that does not seem to fit this context, which is talking about action spaces.)
- Section 4.2.1: The paper completely lost me here; the clarity needs improvement.  Some examples of things I am confused about from this section:
    - Comparing Figure 2 to Algorithm 1, the phases and terminology are different.  Is the “Collecting Phase” the same thing as “Stage 1”?  If so, another issue is that Figure 2 implies that data is only collected at this stage, but Algorithm 1 talks about using that data to train during this stage.
    - Figure 2: Shouldn’t there be an arrow from the k vector to the code table?
    - It is not clear (from the figure, the caption, or the text) where the collected data is going or how it is being used in the training phase (or even if it is being used in the training phase).
    - “All necessary data will be packed into a transition and put into buffers.”  I am not sure what a transition means in this context (maybe a {s, a, s’, r} tuple?), or where the buffers go, or how the data is subdivided into different buffers
    - It’s not clear to me how the decoder, the hybrid action space, and the selected action (a_t) interact.  (Although this could be related to the clarity issues from Section 3, rather than a problem with Section 4.2.1.)
    - Does the collecting phase happen once (many episodes of data), and then the training phase happen after?  Or does the algorithm loop back and forth between the phases? (And if so, how often?  Once per episodes?  Once per timestep?  Some other interval?)  The answers to these questions are not clear from Figure 2 and Section 4.2.1.  Also, while Algorithm 1 may give answers to these questions, the relationships between the components of Algorithm 1 and the components of Figure 2 is not clear (discussed more above).
- Claim: “ADQ significantly outperforms DQN with manually discretized action space”. In RL, 5 seeds is not usually sufficient to make a claim like this, due to the large variance between runs.  This claim is not convincing, nor are several other claims in the paper based on a similarly-small number of seeds.

**Question:** “the decoder uses the code vector e^d”: should this be z^d?


**Summary Of The Paper:**

The authors propose NDRL, which is a class of methods to automatically discretize action spaces.  They identify issues with prior work and with their methods, and propose improvements to address these issues.  They analyze their methods empirically.

**Summary Of The Review:**

The authors study an important and interesting problem; however, the clarity of the paper could be improved.  There are also some empirical problems.  However, the empirical contribution overall was strong.

---

> ### Author Response · Authors · 2022-11-19
> **Response to Reviewer f3BE**
>
> Thanks for your feedback and advice, we will address your concerns and questions as follows:
>
> > Question about writing:
>
> For the definition of Hybrid Action Space and VQ-VAE, we will add revelant definitions and descriptions for both mathematical notation and detailed design. The decoder indeed uses the code vector z^d rather than e^d. Ths revision version of paper will be updated before the end of november.
>
> > Question about Figure1:
>
> In Figure 1, we calculate the shannon entropy $H(x) = -\sum p(x)log_ep(x)$ for different action spaces, and x is the random variable that represents which discrete action choice the action belongs to. Figure 1a is the original continous action space so we will modify related captions. x and y is the two-dimension coordinates of LunarLander game, i.e., horizontal direction and vertical direction. For the sentence about “HandCrafted is obtained...", it describes the internal mechanism of Lunarlander game, which is the physcial simulation rules of this game, so we can use this rule to design the hand-crafted action space.
>
> > Question about section 4.2.1:
>
> We will improve the clarity of section 4.2.1 and related captions in the modified paper. Stage 1 and Stage 2 is to describe the pre-training and normal training stage. Inside each stage, there are the alternate executions between collecting phase and training phase. Transition is the training data tuple, e.g. <s,a,s',r,done> for DQN. When collecting `n_sample (usually use 80)` transition data, they will be saved in two different buffers, then the training phase starts with sample data from corresponding buffer to execute training. After training `update_per_collect (usually use 10)` iterations, the pipeline will back to execute collecting procedure.
>
>
>
>
> > Claim: “ADQ significantly outperforms DQN with manually discretized action space”. In RL, 5 seeds is not usually sufficient to make a claim like this, due to the large variance between runs. This claim is not convincing, nor are several other claims in the paper based on a similarly-small number of seeds.
>
> We argue with this comments because many classical deep reinforcement learning papers, like [1], [2], [3] also reported five random seed results. And we have drawn the standard variation range in all of our training curves, it is obvious to show that ADQ exhibits lower variance than DQN with manually discretized action space among different environments.
>
> > Question about reproducibility:
>
> We will release the main code implementations in anonymous github before the end of november.
>
> [1]. https://arxiv.org/pdf/1509.02971.pdf
>
> [2]. https://arxiv.org/pdf/1707.06347.pdf
>
> [3]. https://arxiv.org/pdf/1801.01290.pdf

---

### Decision · Program_Chairs · 2023-01-20

**Decision:**

Reject

**Justification For Why Not Higher Score:**

There were major questions left about clarity of the paper, proper baseline comparisons and explaining all the design choices made in experiments.

I believe many questions are left unanswered. For instance, details about the training setting, architecture and baselines (e.g. MuJoCo-Gym-v3's Humanoid task referred to by one of the reviewers and its comparison to BDQ). The paper needs to be stronger on all fronts (clarity, experiments, writing, explaining decisions) before it is ready for publication.

**Justification For Why Not Lower Score:**

N/A

**Metareview: Summary, Strengths And Weaknesses:**

This paper studies the topic of building a unified action representations consisting of mixed continuous and discrete action spaces. Traditionally these hybrid action spaces required domain knowledge and hacks but the authors investigate a VQ-VAE based approach to represent any action type via a vector quantized encoder-decoder style architecture -- bringing all actions into a unified representation.

Since the semantics of action is learned over the course of training, the algorithm operates in two iterative phases: (1) learn the latent action space and then (2) use RL on top of the learned action space.

I think this is largely an open problem in RL, especially when its applied to the real world where the action spaces are large and hybrid. And this paper formulates the problem in an interesting manner that has the potential to scale well for many real world problems.

However, the general consensus is that the paper needs more work before it is ready for a publication. Even after the rebuttal process, there were major questions left about clarity of the paper, proper baseline comparisons and explaining all the design choices made in experiments. I believe that this paper is too early in its life cycle and I would urge the authors to resubmit this paper with all the feedback received in this process -- I do believe that this is a promising solution to an important problem.

**Summary Of Ac-Reviewer Meeting:**

N/A